

# Characterizing programmed cell death features in osteoarthritis through integrative multiomics and machine learning analysis

Qinchao Sun, Ye Zhong, Gaoxiang Huang and Yongpei Lin

Department of Orthopedics, Hangzhou Fuyang First People's Hospital, Hangzhou, China

## ABSTRACT

**Background**. Programmed cell death (PCD) is an essential biological process in maintaining tissue homeostasis and eliminating damaged or unnecessary cells. Signaling molecules profoundly affect cellular metabolism and are crucial in various diseases; however, their role in osteoarthritis (OA) remains unclear. This study aimed to systematically evaluate the predictive value, genetic alterations, and therapeutic implications of PCD-associated genes in OA.

**Methods**. We performed multiomics analyses, integrating transcriptomic and single-cell transcriptome data. The biological importance of PCD genes was investigated using differential expression analysis, functional enrichment analysis, pathway analysis, weighted gene co-expression network analysis, and many machine learning models. Additionally, we evaluated diagnostic efficacy, immune infiltration, and competing endogenous RNA networks associated with these genes. We established an *in vitro* OA model using hypoxic treatment of ATDC5 chondrocyte cells and conducted extensive research on the expression and function of key PCD-related genes.

**Results**. The key PCD gene was identified as markedly dysregulated in OA. Elevated expression of S100A9, PMAIP1, and EDA2R was observed in OA samples, indicating these genes as potential risk factors for OA. However, FASN expression was reduced in OA samples compared to the normal group, indicating its potential role as a protective gene in OA. Furthermore, PCD emerged as a reliable diagnostic marker with improved predictive accuracy. Functional experimental studies demonstrated that S100A9, PMAIP1, and EDA2R downregulation through small interfering RNA, alongside FASN gene overexpression through plasmid transfection, significantly ameliorated hypoxia-induced reductions in cell viability, decreased hyaluronan secretion, and increased secretion of inflammatory cytokines (tumor necrosis factor-alpha and interleukin-6).

**Conclusion**. Utilizing a multi-model synergistic artificial intelligence framework, we demonstrated the remarkable potential of PCD to provide individualized vulnerability assessments and customized recommendations for metabolic and immunotherapeutic interventions in OA. We identified abnormal expression of four hub genes associated with PCD and examined their biological functions, thereby facilitating new avenues for research into the role of PCD in OA and other immune-mediated diseases.

Corresponding author
Yongpei Lin, linyong-pei2025@163.com

## INTRODUCTION

Osteoarthritis (OA) is the most common joint disorder globally, affecting approximately 10% of men and 18% of women over the age of 60 (*Abramoff & Caldera, 2020*). The resultant pain and functional impairment can be exceedingly debilitating. In developed countries, the socio-economic impact is significant, accounting for 10%–25% of the gross domestic product (*Glyn-Jones et al., 2015*). OA is the leading cause of disability among the elderly, resulting in pain, functional decline, and diminished quality of life. Current treatment options include low-impact aerobic exercise, weight reduction, acupuncture, glucosamine and chondroitin sulfate supplementation, and surgical interventions (*Jiang, 2022*). The insufficient understanding of the molecular mechanisms underlying OA pathogenesis hinders the formulation of effective strategies to halt OA progression or avert irreversible cartilage degradation, apart from total joint replacement (*Xia et al., 2014*). To better present the recent existing research on mechanisms of OA, we have compiled a literature table (Table S1).

The delicate balance of metabolic activities in articular cartilage is disrupted by genetic aberrations in the TGF-β/Smad, Wnt/β-catenin, and Ihh gene regulatory networks, leading to irreversible degradation of the extracellular matrix. Recent mouse models of OA have highlighted the upregulation of catabolic enzymes, including MMP-13 and ADAMTS5, suggesting their potential as therapeutic targets for modulating OA progression (*Marshall et al., 2018*). These factors can be taxonomically classified into mechanical impacts, aging effects, and genetic factors (*Xia et al., 2014*). Numerous studies have clarified the molecular mechanisms underlying OA pathogenesis. Chondrocytes in joints are essential regulators of articular cartilage homeostasis, preserving its structural and functional integrity (*Takahata et al., 2021*). Recent studies have indicated that the equilibrium of articular chondrocytes can be disrupted by various factors, including abnormal mechanical forces and aging (*Fang et al., 2021*). Furthermore, genetic mutations in the TGF-β/Smad, Wnt/β-catenin, and Ihh gene regulatory networks disrupt the metabolic processes in articular cartilage, leading to the irreversible degradation of the extracellular matrix. Recent murine models of OA have demonstrated the upregulation of catabolic enzymes, including MMP-13 and ADAMTS5, indicating their potential as therapeutic targets for modulating OA progression (*Bernabei et al., 2023*). Manipulating these molecular entities within articular chondrocytes may impact the regeneration of articular cartilage.

Programmed cell death (PCD) is an essential biological process in maintaining tissue homeostasis and eliminating damaged or superfluous cells (*D'Arcy, 2019*; *Kari et al., 2022*; *Kopeina & Zhivotovsky, 2022*). PCD is a complex phenomenon that occurs through various mechanisms, including apoptosis, anoikis, autophagy, alkaliptosis, cuproptosis, entosis, entotic cell death, immunogenic cell death, ferroptosis, lysosome-dependent cell death, methuosis, necroptosis, NETosis, oxeiptosis, pyroptosis, parthanatos, and paraptosis (*Tower, 2015*). PCD can be conceptualized as an intrinsic maintenance mechanism within our biological systems. Similar to the meticulous organization of our homes to maintain cleanliness and harmony, PCD performs a cellular cleansing function that eradicates damaged or unnecessary cells, thereby enhancing tissue health.

Apoptosis is a widely recognized PCD mechanism essential in maintaining tissue homeostasis and eliminating damaged or excess cells. This highly regulated cellular process is characterized by a series of biochemical and morphological alterations (*Jeong & Seol, 2008*). Pyroptosis, a distinct form of PCD, is initiated by inflammasome activation and caspase-1 proteolysis, leading to cellular swelling, membrane permeabilization, and the release of pro-inflammatory cytokines (*Yu et al., 2021*). Ferroptosis, a new form of PCD, is characterized by iron-dependent cell death and lipid peroxidation (*Jiang, Stockwell & Conrad, 2021*). Autophagy, a conserved cellular process, maintains cellular homeostasis by degrading damaged proteins and organelles, thereby facilitating cellular survival or apoptosis, depending on the context (*Glick, Barth & Macleod, 2010*). Necroptosis, which resembles necrosis-like cell death and is mediated by RIPK1 and RIPK3 activation, is another form of PCD (*Zamzami et al., 1997*). Cuproptosis is a PCD modality induced by copper overload, marked by lipid peroxidation and mitochondrial dysfunction (*Xie et al., 2023*). Entotic cell death occurs exclusively within viable cells and their surrounding regions, distinguishing itself from conventional apoptotic mechanisms (*Tang et al., 2019*). NETosis, induced by the release of neutrophil extracellular traps, represents another form of PCD activated by infections or injuries (*Tang et al., 2019*). Parthanatos is a meticulously regulated PCD process induced by excessive PARP-1 nuclease activation (*Zheng et al., 2022*). Lysosome-mediated cell death entails the translocation of hydrolases into the cytosol through membrane permeabilization (*Mahapatra et al., 2021*). The novel PCD paradigm, alkaliptosis, is induced by intracellular alkalinization (*Chen et al., 2023*), while oxeiptosis utilizes KEAP1's reactive oxygen sensing abilities and interacts with other cell death mechanisms (*Scaturro & Pichlmair, 2019*).

This study aimed to identify clusters closely associated with PCD by analyzing genes intricately implicated in this pathway, each exhibiting unique prognostic and immunological characteristics. Utilizing the distinct features of PCD, we developed a prognostic model for OA. The developed model identified unique prognostic, mutational, and immunological signatures, offering new insights into the molecular mechanisms underlying OA and indicating potential therapeutic targets for further investigation.

## METHODS

### Data collection

Clinical data and transcriptome profiles of patients with OA were obtained from the Gene Expression Omnibus (GEO, http://www.ncbi.nlm.nih.gov/geo) repositories. The analytical cohort comprised 88 samples, with 50 samples carefully selected from GSE89408 and 38 samples from GSE114007. We employed the "combat" function from the "sva" package to mitigate batch effects and improve dataset comparability. Before analysis, all data underwent log transformation. Furthermore, we employed single-cell transcriptome data from the GSE152805 dataset in the GEO database, which includes single-cell transcriptomic profiles for six OA samples, thereby enhancing the scope of our investigation.

### Identification of PCD- related differential genes

Utilizing the comprehensive scholarly resource provided by *Zou et al. (2022)*, we compiled a collection of 18 distinct modalities of PCD and their responding regulatory genetic elements. This collection comprises 580 apoptotic genes, 367 genes implicated in autophagy, seven genes related to alkaliptosis, 338 genes associated with anoikis, 19 genes linked to cuproptosis, 15 genes connected to enteric cell death, 87 ferroptotic genes, 34 genes involved in immunogenic cell death, 220 genes related to lysosome-mediated cell death, 101 necroptotic genes, eight genes associated with netotic death, 24 genes related to NETosis, five genes linked to oxeiptosis, 52 pyroptotic genes, nine genes associated with parthanatos, and 66 genes relevant to paraptosis (Table S2). Additionally, our genetic repository was augmented by eight methuotic genes and 23 entotic genes, culminating in 1,964 genes intrinsically associated with PCD. After eliminating 416 redundant entries, we retained a refined set of 1,548 genes, which are essential for our investigation into the PCD mechanism. We utilized the analytical capabilities of the "Limma" package to determine the extent of differential expression between OA and non-pathological samples. We selected genes exhibiting significant expression alterations by applying criteria of a $\log_2$ fold change ($\log_2$FC) > 1 and a false discovery rate (FDR) < 0.05 (*Nie et al., 2023*). Subsequently, we utilized the "VennDiagram" tool to visually depict the overlap between differentially expressed genes (DEGs) and those associated with PCD.

### Biological function and pathway enrichment analysis

In our analysis of genes exhibiting significant expression disparities between the highest and lowest Programmed Cell Death Index (PCDI) classes, we adhered to thresholds of FDR < 0.05 and $\log_2$ fold change >1. Moreover, to investigate the biological functions and pathways associated with PCDI, we applied gene ontology (GO) and Kyoto encyclopedia of genes and genomes (KEGG) analyses using the "clusterProfiler" software. The previously identified differential genes were utilized as our input data, which we transformed into Entrez identifiers before conducting GO and KEGG enrichment analyses. An adjusted *p*-value threshold of <0.05 was utilized as a criterion for significance during this analytical process.

### Single sample gene set enrichment analysis (ssGSEA)

As an advanced variant of gene set enrichment analysis (GSEA), a single sample gene set enrichment analysis (ssGSEA) enables researchers to provide an enrichment score to each individual sample within a dataset. This advanced technique allows for a more comprehensive analysis of gene expression data, facilitating the identification of subtle patterns and relationships that may not be detected using conventional GSEA. The enrichment score reflects the degree to which a gene set is enriched in a given sample. Herein, we utilized the ssGSEA approach to determine the concentration fraction of PCDs in OA samples.

### Single-cell transcriptomic analysis

We employed the "Seurat" R package to examine cellular heterogeneity in OA samples, utilizing single-cell RNA sequencing (scRNA-seq) data. To enhance the integrity of our

analysis, we carefully eliminated low-quality cells. These included cells with gene expression identified in three or fewer cells, those with fewer than 200 detectable genes, cells with a mitochondrial gene ratio above 15%, a ribosomal gene ratio below 3, and those with nFeature counts exceeding 7,500 (*Pu et al., 2024*; *Schneider et al., 2021*). The raw count data was normalized using Seurat's "LogNormalize" method with a scale factor of 10,000. Cell clusters were delineated using canonical markers, while dimensionality reduction techniques, including t-distributed Stochastic Neighbor Embedding (t-SNE) and Uniform Manifold Approximation and Projection (UMAP), facilitated clear visualization of these clusters. Gene expression normalization in the core cells was accomplished through a linear regression model, after which the top 3,000 highly variable genes were selected using analysis of variance. Principal component analysis (PCA) was applied to the single-cell samples, leading to the selection of the first 30 principal components (PCs) for subsequent study. The UMAP algorithm was utilized to facilitate the dimensionality reduction analysis of the first 15 PC sample pairs.

The R package "SingleR," supported by the Human Primary Cell Atlas data, Blueprint ENCODE data, and Immune Cell Expression data, was employed as reference data. Marker genes crucial for the manual labeling of distinct clusters were identified using the CellMarker database and previous academic research. The AUCell software (version 1.14.0) was utilized to calculate gene set scores from the scRNA-seq data, with a calibrated threshold to assign activity scores to cells exhibiting PCD. The ggplot2 package (https://CRAN.R-project.org/package=ggplot2) was utilized to generate feature plots.

## Immunosuppression and immune evasion analysis

Employing the ssGSEA method from the GSVA package, we performed a ssGSEA on the identified module genes, focusing on a specific subset of genes associated with immune suppression. This method facilitates the calculation of an immune suppression score for each sample. Subsequently, we utilized the differential analysis tool Limma to evaluate immune suppression scores across various sample groups, thereby identifying significantly divergent immune suppression states. To illustrate these findings, we employed graphical tools, including ggplot2 and boxplot, to display the immune suppression scores across different groups, and we generated heatmaps to depict the immune suppression characteristics of multiple genes. Additionally, we conducted an immune evasion analysis on the identified model genes by initially assessing their expression differences across distinct groups to evaluate their potential for immune evasion. Subsequently, we utilized the Tumor Immune Dysfunction and Exclusion (TIDE) tool to evaluate the immune evasion potential of the samples. TIDE calculates an immune evasion score for each sample by synthesizing gene expression data with established immune evasion mechanisms. R packages, including ggplot2, were employed to effectively illustrate the outcomes of the immune evasion investigation, highlighting the characteristics of immune evasion and their clinical significance.

## Weighted gene co-expression network analysis (WGCNA) analysis

The R package weighted gene co-expression network analysis (WGCNA) was utilized to investigate genes associated with PCD scores. The goodSamplesGenes function in WGCNA

was initially employed to assess the need for gene filtration and to identify a suitable soft threshold. A gene module comprising at least 30 components was constructed following the framework of the hybrid dynamic tree-cutting algorithm. This facilitated the establishment of a co-expression network, highlighting the complex architecture of biological systems. Subsequently, the Pearson correlation coefficient was utilized to analyze the association between the module eigengene and PCD score. The study resulted in a Venn diagram that depicted the intersection between the association module and DEGs, clarifying the convergence of these genomic entities.

## Development of diagnosis model

Our comprehensive analysis incorporated ten independent machine learning techniques, yielding an extensive evaluation of 101 unique algorithmic permutations. The ensemble comprised various well-regarded approaches: support vector machines, least absolute shrinkage and selection operator, gradient boosting machine, random forest, elastic net, stepwise Cox, k-nearest neighbors, extreme gradient boosting, and ridge regression. Predictive models were developed utilizing the GSE89408 dataset and subsequently validated across two independent external datasets (GSE55235 and GSE114007). We assessed model performance using metrics including the area under the curve (AUC) and accuracy index, identifying 23 models proficient in gene screening and importance assessment to identify essential genes. Moreover, we developed calibration and receiver operating characteristic (ROC) curves to carefully evaluate the prognostic capabilities of the key genes identified.

## Cell culture

The ATDC5 chondrocyte subline was cultured in a medium comprising an equal mixture of Dulbecco's modified Eagle's medium (DMEM) and Ham's F-12 (Gibco, Invitrogen, Carlsbad, USA), supplemented with 5% fetal bovine serum (Gibco), 1% antibiotic-antimycotic solution (Beyotime, Shanghai, China), 10 µg/mL human transferrin, and 30 mM sodium selenite (Beyotime). The cells were purchased from the American Type Culture Collection (ATCC, Manassas, VA, USA). An *in vitro* OA model was developed by subjecting ATDC5 chondrocytes to hypoxic conditions (*Sun et al., 2018*), specifically by incubating them in an environment with 5% $O_2$ at 37 °C for 48 h.

## Cell viability

Cell viability was evaluated using the Cell Counting Kit-8 (CCK-8) assay (Beyotime). Cells were inoculated into 96-well plates at a density of 5,000 cells per well. After treatment with various conditions, CCK-8 solution was added to each well, and the plates were incubated for an additional hour in a humidified atmosphere at 37 °C with 5% $CO_2$. Optical density was subsequently measured at a wavelength of 470 nm.

## Enzyme-linked Immunosorbent Assay (ELISA)

An enzyme-linked immunosorbent assay (ELISA) was utilized to evaluate the concentrations of hyaluronan, tumor necrosis factor-alpha (TNF-α), and interleukin (IL)-6 in the culture medium of ATDC5 cells under different treatment conditions. ATDC5 cells

**Table 1** Primer sequences of qRT-PCR.

| Gene | Forward primer | Reverse primer |
|---|---|---|
| FASN | GGAGGTGGTGATAGCCGGTAT | TGGGTAATCCATAGAGCCCAG |
| S100A9 | ATACTCTAGGAAGGAAGGACACC | TCCATGATGTCATTTATGAGGGC |
| PMAIP1 | GCAGAGCTACCACCTGAGTTC | CTTTTGCGACTTCCCAGGCA |
| EDA2R | CACACTGCATAGTCTGCCCTC | GCCTTCTGGACCCGATTGA |
| β-actin | GGCTGTATTCCCCTCCATCG | CCAGTTGGTAACAATGCCATGT |

were initially organized into groups and treated according to the experimental protocol. Subsequently, culture medium from each group was collected, and ELISA kits (Cusabio Biotech, Wuhan, China) were utilized to quantitatively measure the concentrations of hyaluronan, TNF-α, and IL-6, adhering strictly to the manufacturer's instructions.

## Reverse transcription, quantitative real-time PCR (qRT-PCR)

Total RNA was extracted from the treated ATDC5 cells using Trizol reagent (TaKaRa, Shiga, Japan) to investigate the expression profiles of target genes in ATDC5 cells under different treatment conditions. Three samples were obtained from each group. This was followed by the synthesis of complementary DNA (cDNA) through reverse transcription, enabling further gene expression analyses. After synthesizing cDNA using reverse transcription, quantitative real-time polymerase chain reaction (qRT-PCR) was performed using the SYBR Green qRT-PCR quantitation kit (TaKaRa) procedure to amplify and quantify the target genes. The residual cDNA was stored at −20 °C. The relative quantification method ($2^{-\Delta\Delta CT}$) was utilized to accurately assess gene expression levels. By utilizing β-actin as a dependable internal control gene (*Lu et al., 2023*), we effectively mitigated variability among samples, facilitating precise evaluation of target gene expression changes across different treatment conditions. Table 1 presents the primers utilized for qRT-PCR.

## Plasmid overexpression

To induce overexpression of the FASN gene, a cDNA clone plasmid of the FASN gene was introduced into ATDC5 cells using the Lipo3000 transfection reagent. After transfection, the cells were cultured in either 5% $O_2$ to simulate physiological hypoxia or 21% $O_2$ to represent normoxia to evaluate the effects of FASN gene overexpression under different oxygen concentrations. FASN gene expression levels were quantified using qRT-PCR 48 h.

## Small interfering RNA interference

We performed small interfering RNA (siRNA)-mediated knockdown experiments targeting the S100A9, PMAIP1, and EDA2R genes in the ATDC5 cell line to examine their roles in cellular processes. The siRNAs were introduced into ATDC5 cells during their logarithmic growth phase with the Lipo3000 transfection reagent to achieve effective gene silencing. To ensure the reliability and reproducibility of the experimental results, we included negative controls (NC, utilizing nonspecific siRNA) and positive controls (using siRNA with established efficacy). Post-transfection, the cells were cultured under standard conditions. Subsequently, qRT-PCR was employed to measure the expression levels of the S100A9,

PMAIP1, and EDA2R genes 48 h after transfection, thereby verifying the knockdown efficiency of the siRNAs.

## Statistical analyses

Our comprehensive analysis utilized various statistical techniques, which were carefully implemented to evaluate the importance of identified discrepancies and relationships in the research study. To illustrate the variability of values around the mean, data are presented as mean ± standard deviation (SD). We employed Cox regression models and Kaplan–Meier survival analysis to investigate the impact of risk factors on survival outcomes, leveraging their respective strengths for this analysis. Additionally, Pearson correlation analysis was performed to investigate the correlations among variables. All statistical analyses and data visualization were performed using R Studio (version 4.3.1). A $p > 0.05$ was considered the threshold for statistical significance.

## RESULTS

### Preliminary screening of mtPCDI regulators

We commenced our investigation by compiling a collection of essential regulatory genes, including 18 carefully curated patterns of PCD from academic sources (*Zou et al., 2022*). Our objective was to examine the differential expression of PCD-related genes in OA. To achieve this, we performed a thorough analysis of two independent, publicly accessible gene expression datasets: GSE89408 and GSE114007. We effectively eliminated batch effects and generated volcano plots (Figs. 1A–1B). We employed rigorous criteria, establishing thresholds at |log$_2$ fold change|$>$ 1 and an adjusted $p < 0.05$, which identified several genes with significant upregulation (indicated in red) or downregulation (indicated in blue) in the OA environment compared to normal samples (Fig. 1C). Several genes associated with critical PCD processes, including ECT2, MAPK1, and SREBF1, consistently exhibited upregulation, highlighting their possible role in OA. Hierarchical clustering analysis further validated the differential expression patterns, demonstrating a clear distinction between OA and normal samples based on the identified DEGs (Fig. 1D). The heatmaps depict unique gene expression profiles in OA, marked by clusters of genes exhibiting significant up-regulation or down-regulation. This clustering pattern highlights substantial gene expression disparities between the two groups, highlighting the potential biological significance of these DEGs in OA.

### Functional enrichment analysis

KEGG pathway analysis revealed key pathways, including "rheumatoid arthritis," "osteoclast differentiation," "cell adhesion molecules," and "phagosome" (Fig. 1E). These enriched pathways imply that the differential genes are implicated in biological processes, including extracellular matrix dynamics, bone development, growth, repair, reconstruction, and immune modulation, presenting prospective therapeutic targets for OA. The GO enrichment study revealed a compelling narrative, indicating that DEGs in OA had significant implications for essential biological processes. These specifically include "neutrophil activation in immune response," "neutrophil degranulation," and

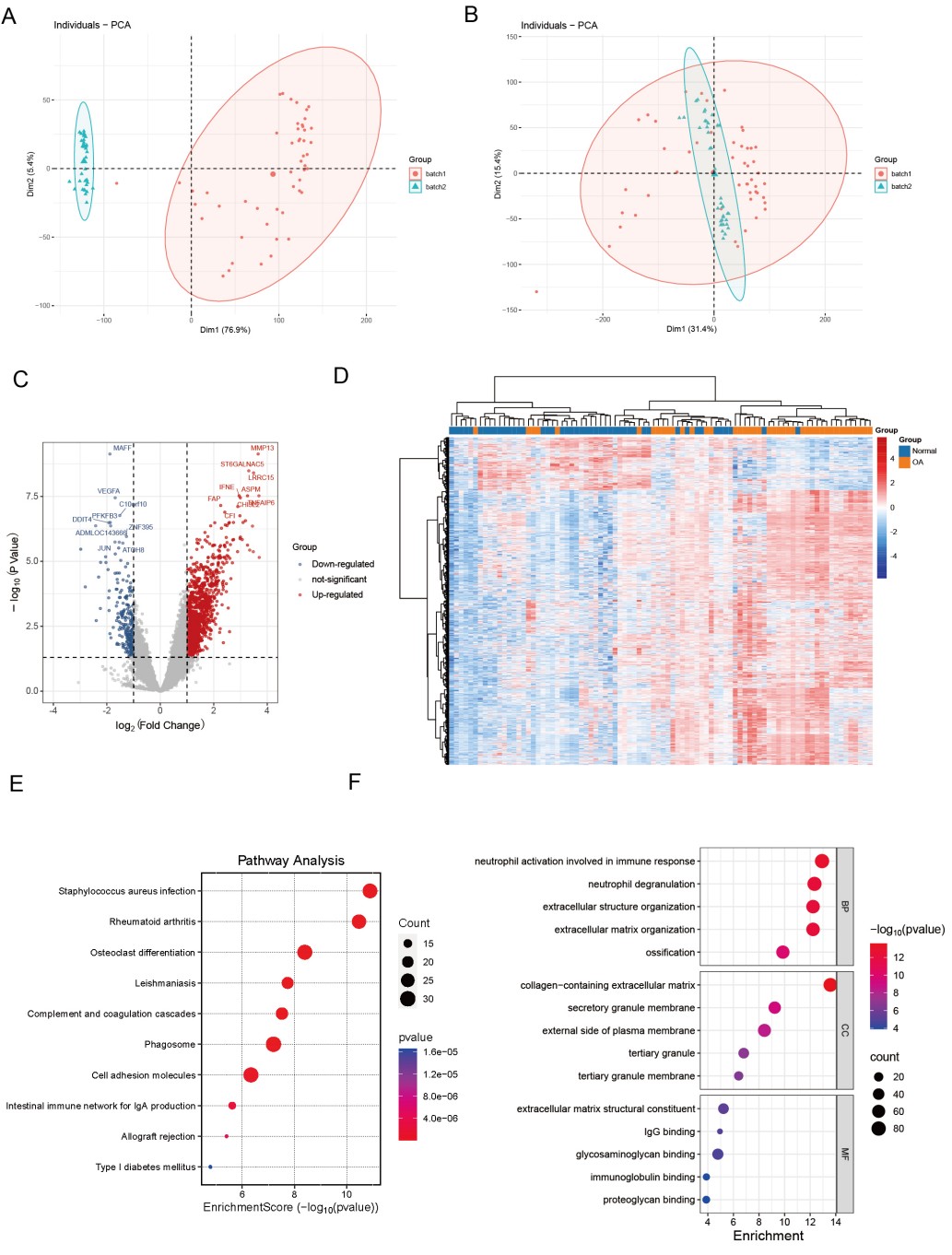

**Figure 1  Differential expression and integration analysis in OA.** (A) PCA plot shows the sample distribution before the removal of batch effect; (B) PCA plot shows the sample distribution after the removal of batch effect; (C) Volcano plots showing DEGs between OA and normal samples in integrated datasets. Upregulated genes are shown in red, downregulated in blue, with key palmitoylation-related genes labeled; (D) Heatmaps displaying hierarchical clustering of OA and normal samples based on DEGs in integrated datasets, revealing distinct expression patterns; (E) KEGG pathway analysis showing enrichment in critical pathways like rheumatoid arthritis, osteoclast differentiation, (continued on next page...)

**Figure 1 (…continued)**
phagosome and cell adhesion molecules; (F) The GO enrichment analysis illuminates a constellation of significantly enriched terms, spanning the triumvirate of biological process, molecular function, and cellular component categories. Salient processes encompass neutrophil activation in immunological response, neutrophil degranulation, extracellular structure orchestration, collagen-laden extracellular matrix, and immunoglobulin tethering. These pivotal elements underscore the intricate tapestry of biological interactions embedded within the data.

"extracellular structure organization." These findings indicate a role in the complex interaction of immune regulation and the dynamic remodeling of the extracellular matrix (Fig. 1F).

## Construction of PCD scores

Hierarchical cluster analysis confirmed the varying expression patterns of specific PCD genes, demonstrating a clear distinction between OA and normal samples (Fig. 2A). Subsequent ssGSEA based on 18 PCD genes revealed significant differences between OA and normal samples across various scores, including those for apoptosis, autophagy, and necroptosis. After detecting these distinct gene expression patterns, we systematically classified them into seven distinct PCD categories: NETosis, lysosome-dependent cell death, oxeiptosis, NETosis, immunogenic cell death, anoikis, and entosis (Fig. 2B). To clarify the relationship between DEGs and PCD, we performed an enrichment analysis focusing on their intersection. The KEGG analysis revealed pathways significantly enriched in lysosome function, apoptosis, and rheumatoid arthritis (Fig. 2C), while the GO enrichment analysis highlighted genes abundant in pathways associated with the regulation of apoptotic signaling, positive modulation of apoptotic signaling, and other aspects of cell death. These findings offer significant insights into the complex molecular mechanisms underlying OA pathophysiology, specifically focusing on neutrophil degranulation—processes closely associated with PCD and immunological regulation (Fig. 2D).

## Single-cell transcriptomic analysis

Our preliminary study involved the careful selection of resilient, non-degradable cells, resulting in a substantial collection of 26,131 core cells for further analysis. Subsequently, we performed a genetic variance analysis on these core cells, identifying an amazing 3,000 genes exhibiting high variability. PCA was performed on five single-cell specimens, demonstrating a coherent distribution of the samples. We identified 30 PCs with $p > 0.05$ for further investigation. The robust t-SNE algorithm effectively categorized the core cells into 16 distinct clusters (Fig. 3A). We employed the analytical functions of the "singleR" software package and the CellMarker database to discover marker genes. This study enabled the identification of various cell clusters, dividing them into seven separate groups: HomC, RepC, FC, preFC, preHTC, HTC, and RegC (Fig. 3B). We constructed scatter plots to effectively depict the expression patterns of marker genes across different cell types (Fig. 3C) and utilized bubble maps (Fig. 3D) to reveal the expression profiles of key marker genes within each cell type. These visual tools provided significant insights into the genetic markers that define the diverse cell populations. Additionally, we examined the expression of marker genes across various cell types, confirming that each selected marker gene

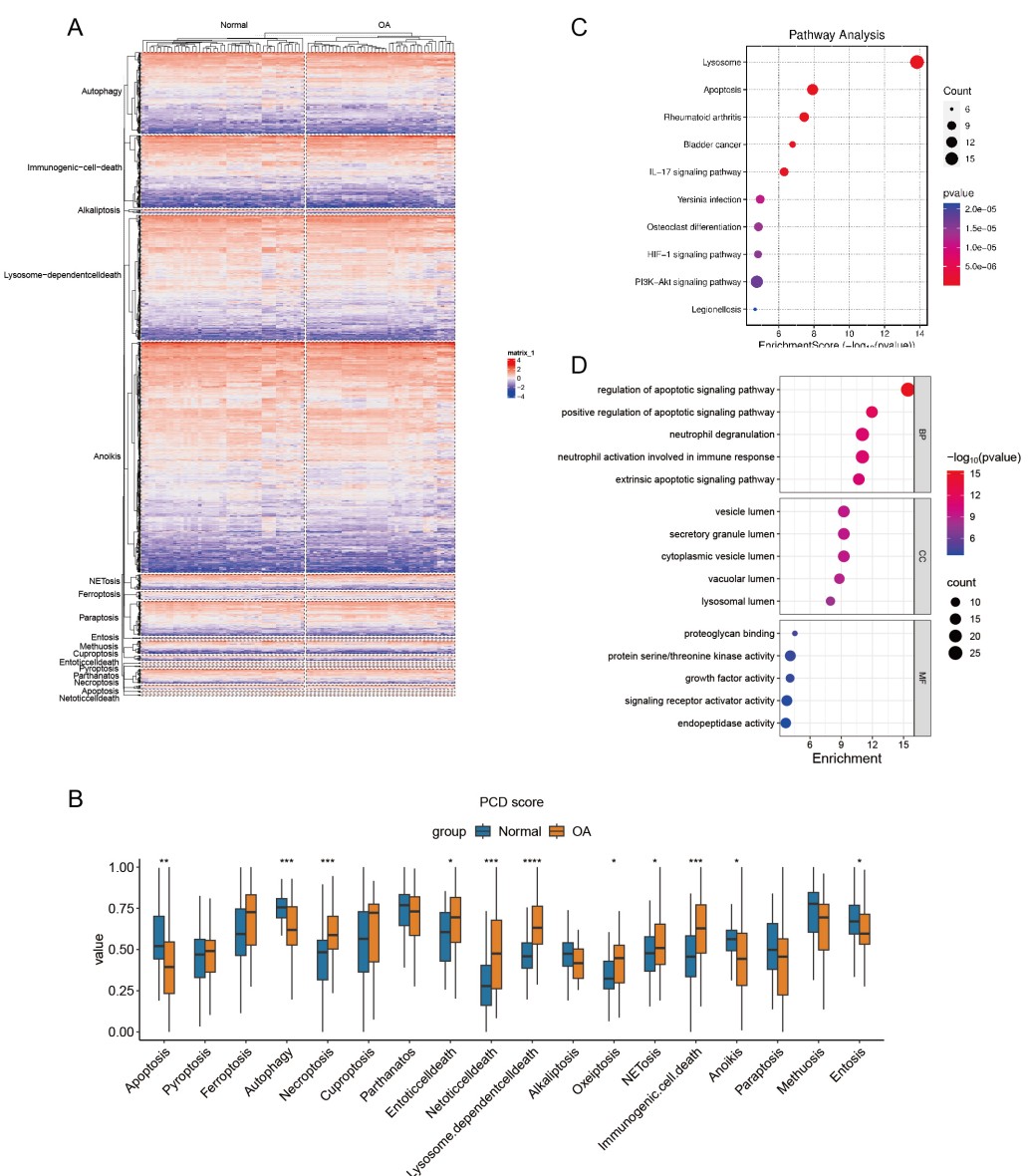

**Figure 2  Differential expression of PCD related genes in OA.** (A) Heatmaps displaying hierarchical clustering of OA and normal samples based on PCD related genes in integrated datasets, revealing distinct expression) Boxplot showing PCD scores between OA and normal samples in integrated datasets based ss-GSEA; (C–D) Dotplots show the KEGG and GO enrichment results of the intersection genes of DEG and PCD.

exhibited high expression in specific cells (Fig. 3E). This finding further supports the reliability of our cell type analyses.

## PCD-related pathway score

We utilized the AUCell R package to evaluate the significance of PCD within each cellular subtype. This tool enabled us to assess the activity of the PCD pathway within each cell subpopulation. This study employed the AUC value as an indicator of pathway activity, with

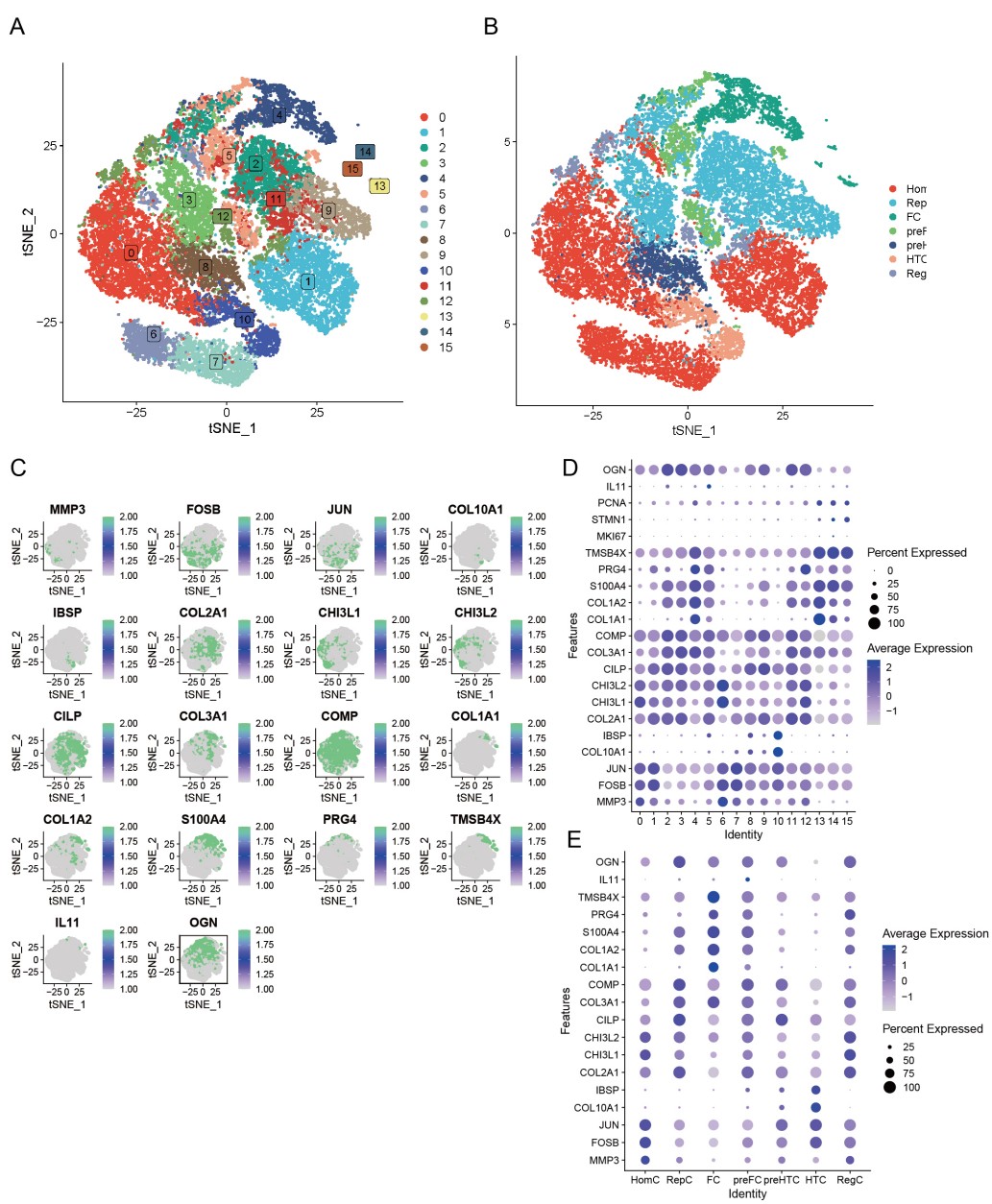

**Figure 3** **Seven unique cell clusters derived from scRNA-seq data, highlighting substantial cellular heterogeneity within OA.** Each cluster possesses distinct annotations, enhancing our understanding of OA pathophysiology. (A) Leveraging the umap algorithm upon the top 30 principal components, we adeptly surmounted dimensionality constraints, culminating in the proficient classification of 16 cellular agglomerations. (B) Harnessing the analytical prowess of singleR and CellMarker, we masterfully ascribed identities to all 7 cellular clusters within the OA milieu, as dictated by the peculiar amalgam of marker genes. (C) t-SNE plots showing the expression of specific genes (MMP3, FOSB, JUN, COL10A1, IBSP, COL2A1, CHI3L1, CHI3L2, CILP, COL3A1, COMP, COL1A1, COL1A2, S100A4, PRG4, TMSB4X, IL11, OGN) across cells. (D) Dot plot displaying the expression of various genes across different cell clusters (0–15). (E) Dot plot illustrating gene expression across distinct cell identities (HomC, RepC, FC, preFC, preHTC, HTC, RegC).

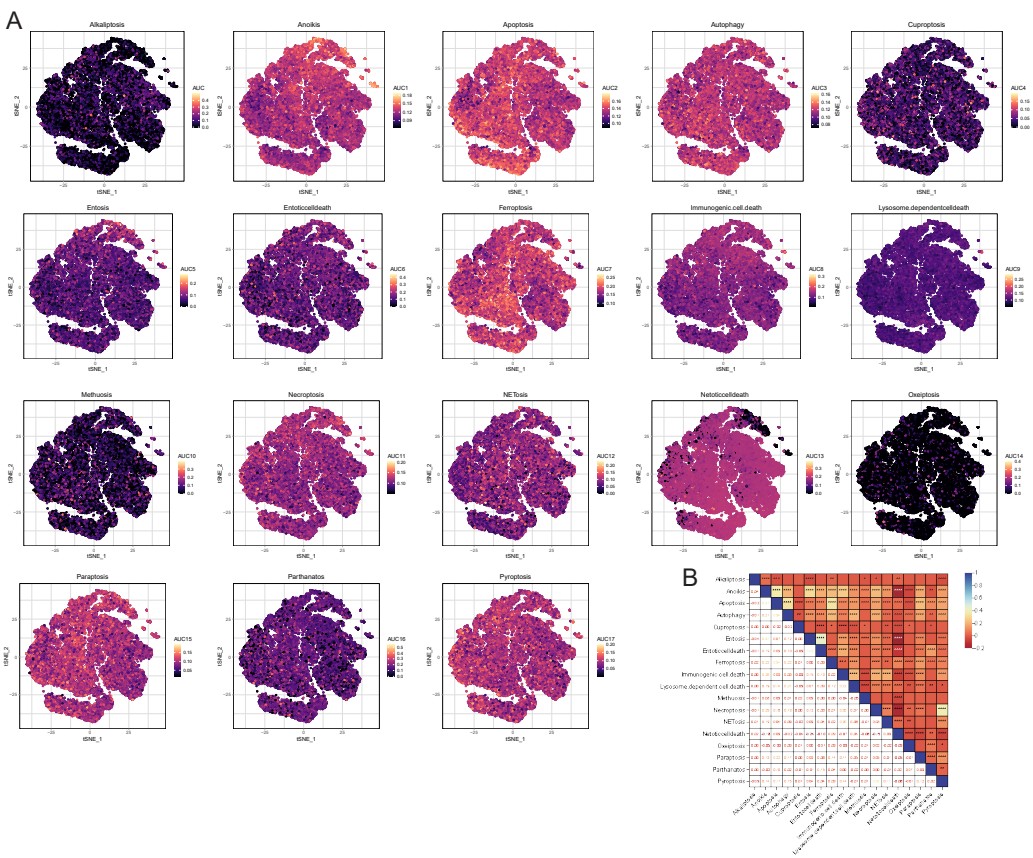

**Figure 4** **AUCell analysis of OA cell subsets.** (A) t-SNE plots illustrating the expression of PCD-related genes within each OA cell subset. (B) Heatmap shows the correlation among the PCD scores based AU-Cell.

higher values signifying greater robustness. We utilized a t-distributed Stochastic Neighbor Embedding (t-SNE) map to depict our findings, which effectively illustrated the dynamic nature of the PCD pathways among different cell subgroups. In this visualization, yellow dots represented regions of increased activity, while gray dots indicated relatively lower activity levels. Notably, pathways including anoikis, apoptosis, autophagy, ferroptosis, necroptosis, NETosis, paratosis, and pyroptosis exhibited high expression levels across different cell clusters (Fig. 4A). Conversely, the expression levels of alternative PCD pathways were low within each cell cluster (Fig. 4A). Consequently, subsequent analyses concentrated on the pathways mentioned above. Besides, a significant correlation was observed among the PCD scores, indicating complex regulatory interactions between these pathways (Fig. 4B).

## Identification of PCD- related key modules

We utilized the WGCNA to identify genes implicated in OA onset and progression. Outlier samples were removed using cluster analysis (Fig. 5A), and the PCD index for each sample is presented in Fig. 5B. During the construction of the co-expression network, we established

the soft threshold power β at 6, corresponding with a scale-free topology fit index of 0.85 (Fig. 5C). The dynamic tree cut algorithm was utilized to merge similar modules by setting the MEDissThres to 0.15, resulting in the identification of 17 modules (Fig. 5D). The modules MEbrown and MEmagenta were identified as key modules based on correlation coefficients and *p* values. MEbrown comprises 2,556 genes, while MEmagenta includes 385 genes (Fig. 5E).

## Unraveling PCD-related genes in OA

We employed various machine learning models focusing on genes related to PCD to achieve a more precise identification of key genes involved in the progression of OA. Initially, we identified the intersection of DEGs, PCD-related genes, and WGCNA modules, resulting in 22 genes selected for further analysis (Fig. 6A). The models derived from these genes demonstrated effective predictive capabilities, with the ridge regression cross-validation 10-fold model (cutoff: 0.75) displaying the highest accuracy and AUC values, averaging 0.713 and 0.725, respectively, across three datasets (Figs. 6B–6C). This diagnostic model, based on PCD-related genes, exhibits strong predictive efficacy. The comprehensive machine learning approach included 23 models for variable selection, and the final key genes were determined by assessing the gene ranking across these models (Figs. 6D–6E). We identified the four principal genes—PMAIP1, EDA2R, S100A9, and FASN—as key contributors.

## Discovery of hub genes for OA

Four genes emerged as central hub genes: S100A9, PMAIP1, EDA2R, and FASN. Their expression levels were significantly elevated in OA compared to controls, except for FASN ($p < 0.05$, Fig. 7A). The ROC curve exhibited high AUC values for these hub genes, indicating their potential as important independent biomarkers for OA (Fig. 7B). These findings collectively highlight the critical roles these hub genes may have in the inflammatory response associated with OA. In the column chart, each feature variable was depicted as a separate score, with their cumulative sum reflecting the probability of OA (Fig. 7C). Focusing on the hub genes, we utilized the miRTarBase version 9.0 database to establish an mRNA-miRNA interaction network (Fig. 7D). Simultaneously, we utilized the TRRUST database to develop an interaction network diagram depicting the interaction between hub genes and transcription factors (Fig. 7E).

## Expression and function of hub genes within an *in vitro* OA model

To establish the OA cell model, ATDC5 chondrocyte cells were cultured in hypoxic conditions for 48 h. The findings indicate that hypoxia exposure significantly inhibited cell proliferation (Fig. 8A), suppressed hyaluronan synthesis (Fig. 8B), and elevated TNF-α and IL-6 levels (Fig. 8C). These findings confirm the successful establishment of an *in vitro* OA cellular model. The analysis of hub gene expression demonstrated consistency with bioinformatics predictions. FASN expression was markedly downregulated in the OA group, while S100A9, PMAIP1, and EDA2R levels were significantly increased, with $p < 0.05$ (Fig. 8D). To further clarify the functions of these hub genes, plasmids were utilized to overexpress the FASN gene, while siRNA was applied to inhibit S100A9, PMAIP1, and

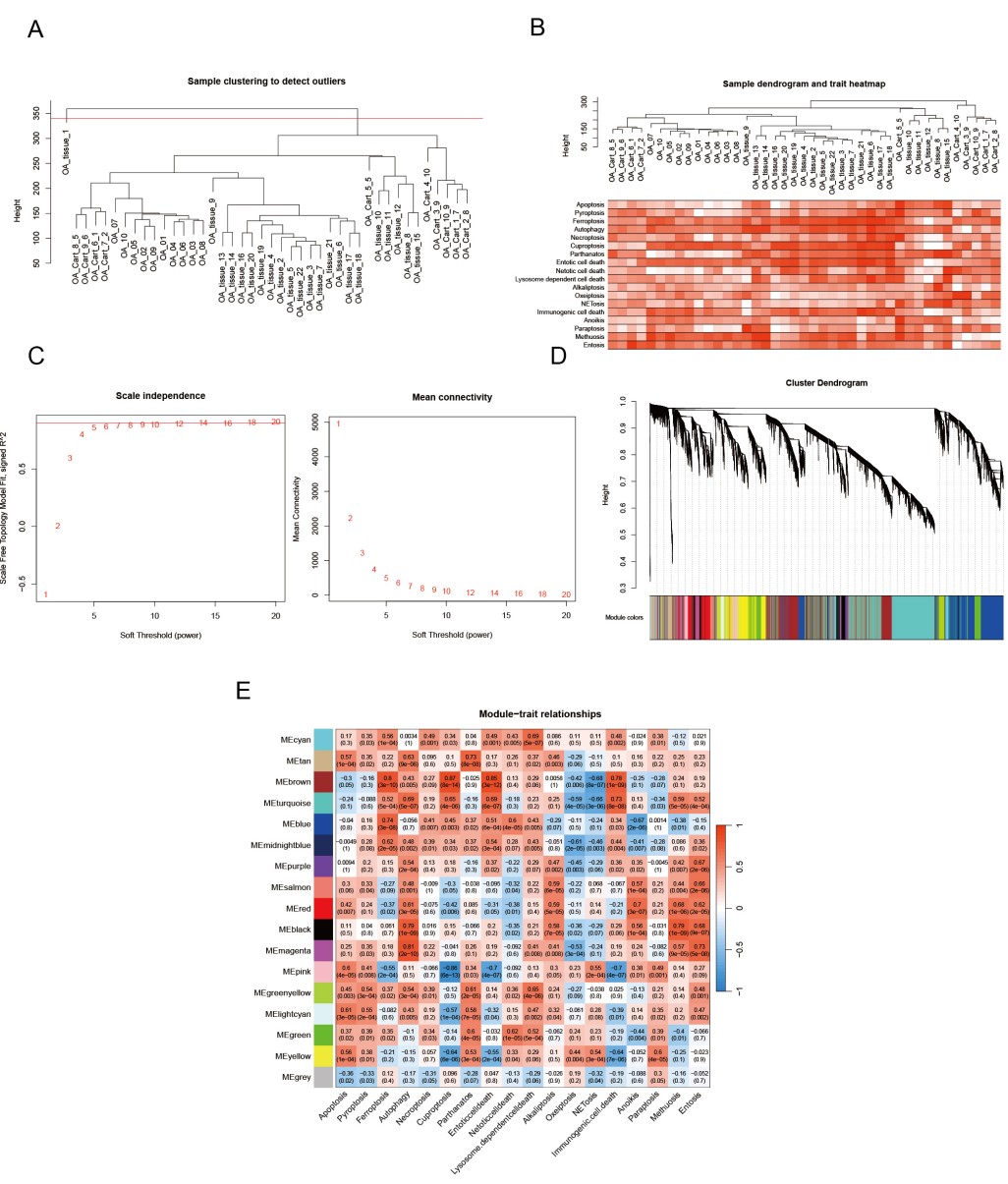

**Figure 5  PCD-related genes are screened by WGCNA.** (A) OA samples cluster tree and remove samples above the red line; (B) PCD pathway scores expression in each sample; (C) Our analytical odyssey commenced with a meticulous dissection of the scale-free index, examining its oscillations across an array of soft-threshold powers (β); (D) The dendrogram (1-TOM) elucidates co-expression network modules, with an ensuing analysis (E) revealing noteworthy associations between these modules and PCD. *P*-values are furnished to corroborate these connections, offering a statistical foundation for the observed relationships and reinforcing the validity of the findings.

EDA2R expression *in vitro* (Fig. 8E). The findings revealed that FASN overexpression and the knockdown of S100A9, PMAIP1, and EDA2R significantly alleviated the hypoxia-induced reduction in cell proliferation (Fig. 8F) and hyaluronan synthesis (Fig. 8G), while an increase in the inflammatory cytokines TNF-α and IL-6 was observed (Figs. 8H–8I) in

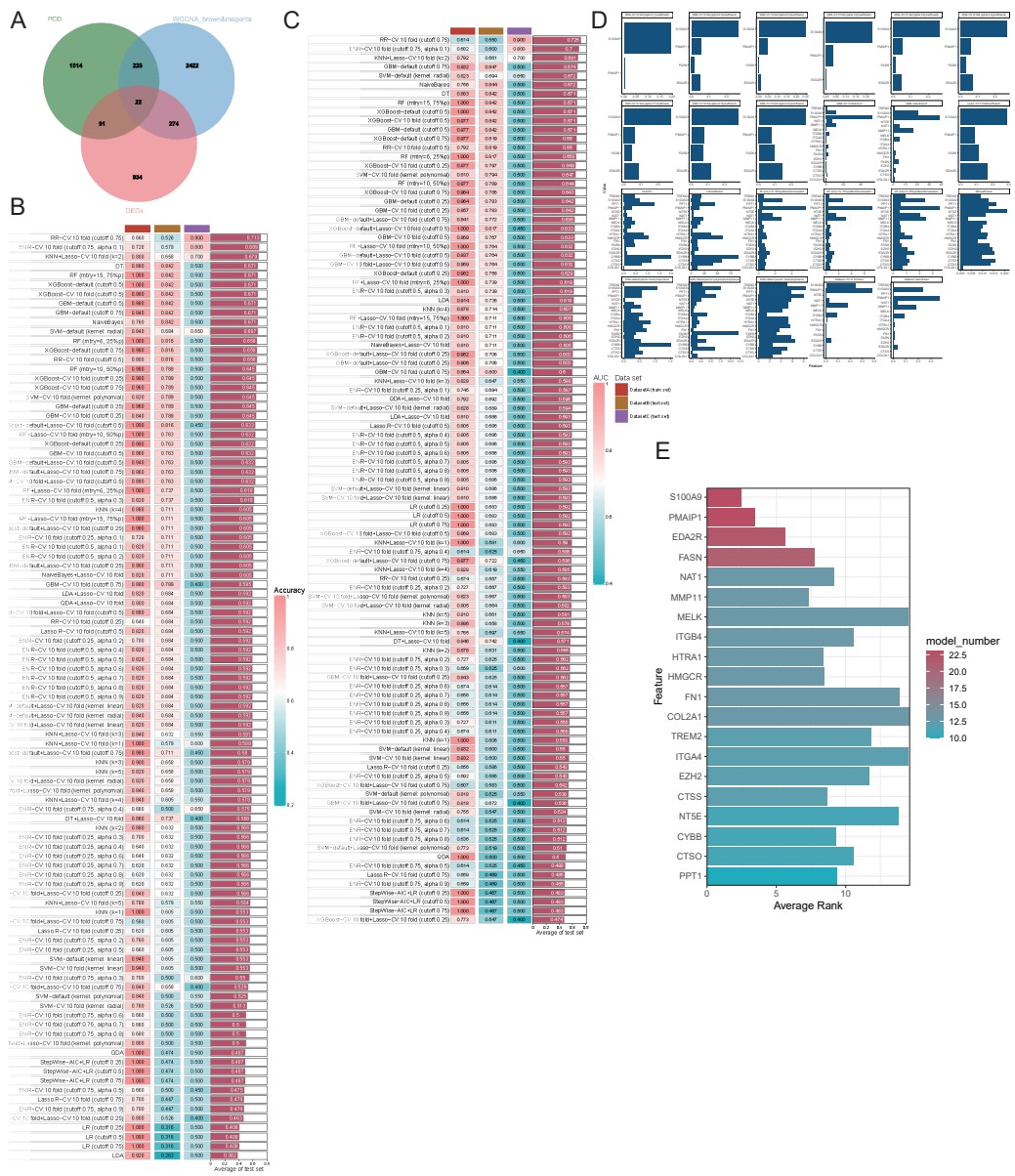

**Figure 6   Evaluating the ability of genes by combining multiple machine learning models.** (A–B) Accuracy of all machine learning models; (C) AUC of all machine learning models; (D-E) Evaluation of the importance of 23 machine learning models with screening capability to PCD-related genes.

ATDC5 cells. These findings highlight the essential functions of these four hub genes in OA onset and progression.

## Immunosuppression and immune evasion analysis

Upon delving into the immunosuppressive-related cells or pathways, a discernible distinction surfaced in Myeloid-Derived Suppressor Cell (MDSC) *Wang et al. (2022)* and fibroblasts Microenvironment Cell Population Counter (MCPcounter) between

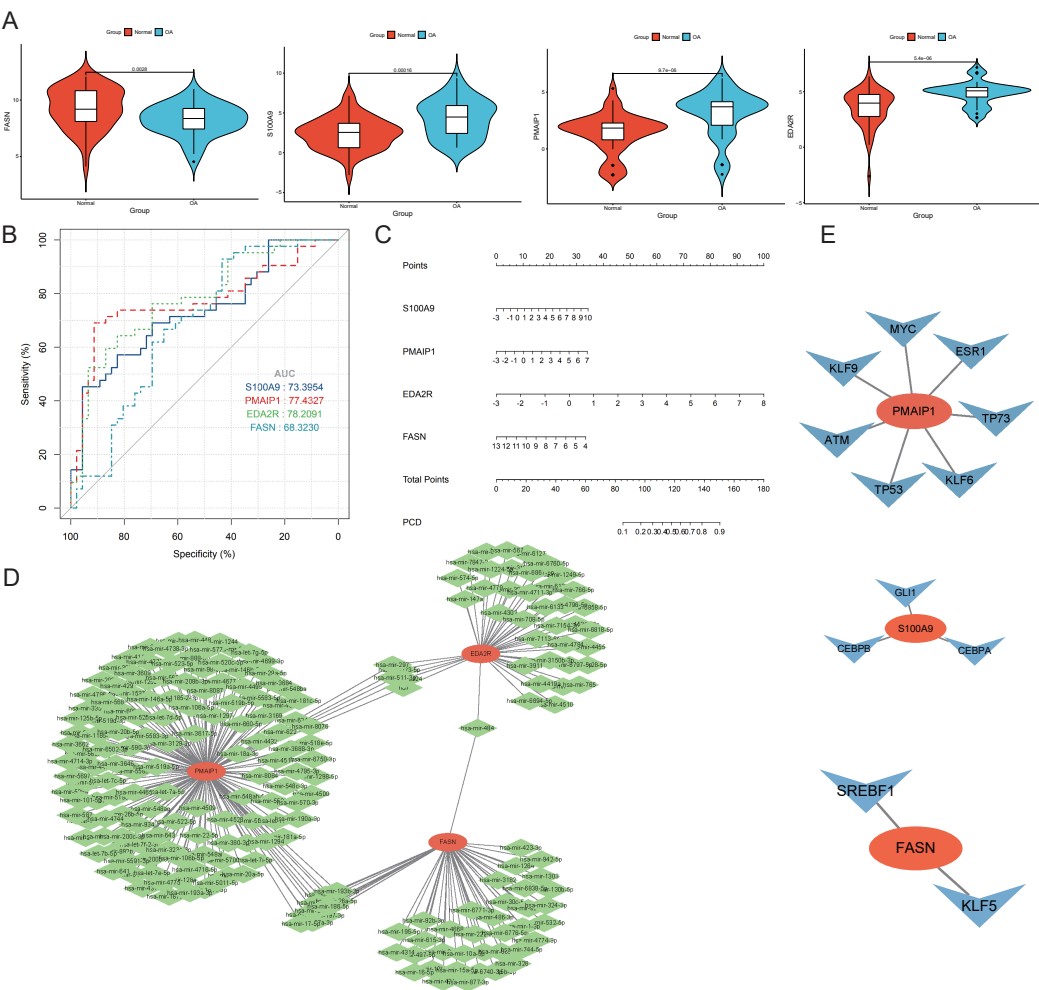

**Figure 7** **Diagnostic performance and molecular function of hub genes.** (A) The hub genes' expression took center stage, illuminating the complex interplay between OA and normal samples. (B) Harnessing the power of Receiver Operating Characteristic (ROC) curves, we meticulously appraised the diagnostic potential of the four distinguished hub genes. The Area Under the Curve (AUC) scores emerged as testament to their performance in discerning OA from normal samples. (C) A column plot wove a tapestry of comprehensive data, unveiling the rich narrative ensconced within its graphical confines. (D) The mRNA-miRNA network took form, a constellation of orange hub mitochondria-related genes and their verdant miRNA counterparts, casting a constellation of complex relationships. (E) The mRNA-TF network unfurled, its orange hub mitochondria-related genes orbiting a galaxy of blue transcription factors, together forging a panorama of interwoven connections.

OA and normal groups ($p < 0.05$), while other features remained statistically static (Fig. 9A). A heatmap offers a comprehensive visual depiction of the significant variations in immunosuppressive-related features, highlighting the contrast between groups with high and low immune infiltration (Fig. 9B). Significant differences were observed in Cancer-Associated Fibroblast (CAF) *Peng et al. (2024)*, fibroblasts MCPcounter, CAFs_EPIC (*Zheng et al., 2021*), macrophages M2 CIBERSORT (*Bao et al., 2021*), and

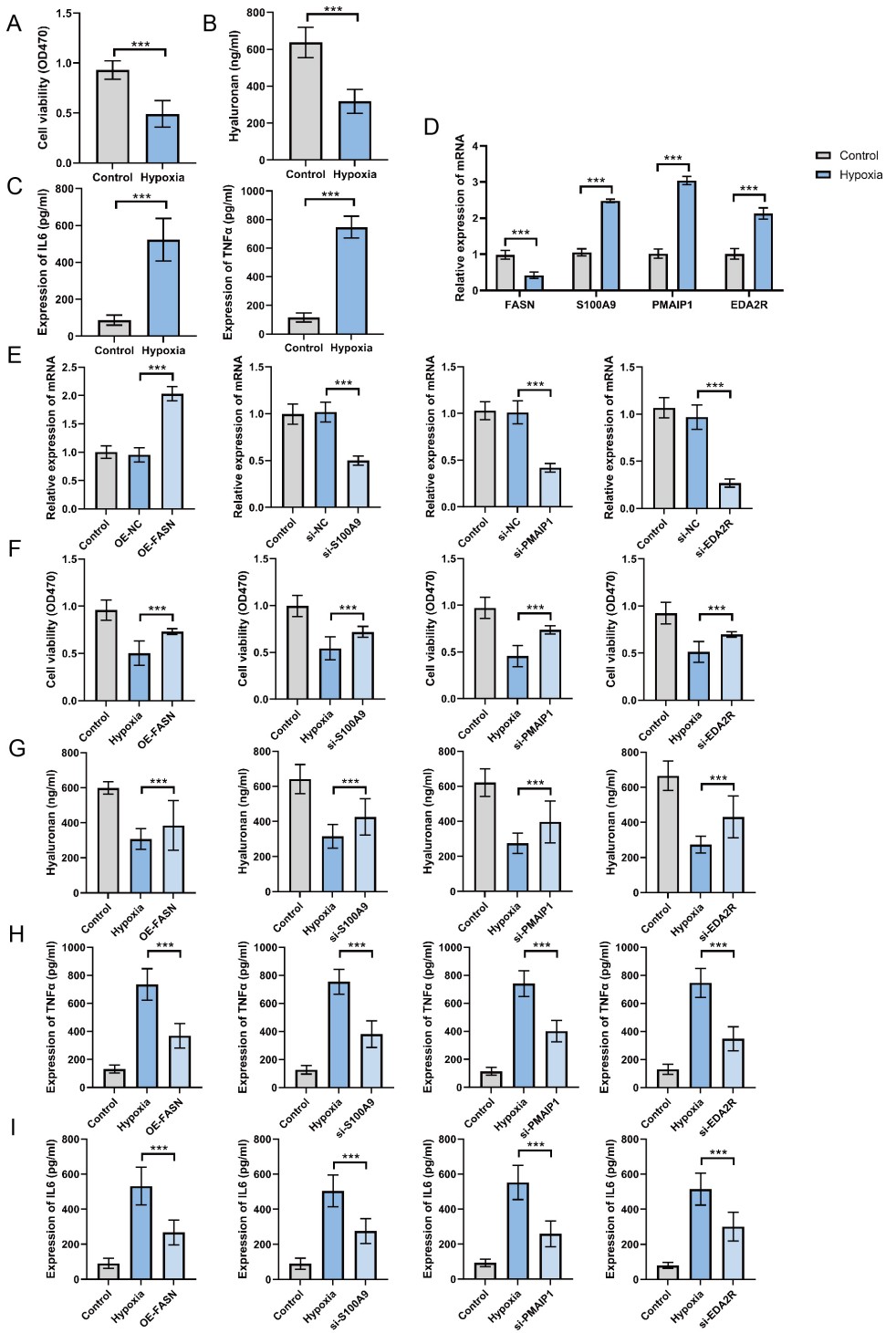

**Figure 8  Expression and function of hub genes within an *in vitro* osteoarthritis model.** (A) Cell survival rates after being subjected to hypoxia. (B) Secretion levels of Hyaluronan after hypoxia treatment. (C) Secretion levels of TNF-α and IL-6 after hypoxia treatment. 

**Figure 8 (…continued)**
(D) To gauge the expression levels of the four central genes under hypoxic conditions, quantitative real-time PCR (qRT-PCR) was employed, offering valuable insights into their responsiveness to this environmental stressor. (E) Additionally, we harnessed qRT-PCR to investigate the expression profiles of these genes in response to plasmid overexpression or siRNA-mediated silencing, thereby illuminating the effects of genetic manipulation on their transcriptional regulation. (F) Detection of cell survival rates after plasmid overexpression or siRNA interference. (G) Detection the occurrence of cell apoptosis after plasmid overexpression or siRNA interference. (H) Secretion levels of TNF-α after plasmid overexpression or siRNA interference. (I) Secretion levels of IL-6 after plasmid overexpression or siRNA interference. ***$p < 0.001$, two-tailed Student's $t$-test was implemented for the examination of (A–D), while a one-way ANOVA formed the basis of our analysis for (E–I).

TAM_Peng_et_al (*Wang et al., 2022*) ($p < 0.05$) when examining immune exclusion-related feature scores (Fig. 9C). A comprehensive heatmap illustrates the variation in immune exclusion-related features between the high and low clusters (Fig. 9D). The Mantel correlation heatmap illustrates a complex network of correlations between immune infiltration and its association with PCD. Multiple immune infiltrations are significantly correlated, indicating a sophisticated regulatory interaction. Furthermore, PCD significantly correlates with various immune infiltrates, including M2 macrophages. The dynamic interaction between PCD and immune mechanisms is highlighted by the significantly different distribution of resting and activated mast cells in response to PCD variations ($p < 0.05$) (Fig. 9E).

# DISCUSSION

The clinical symptoms of OA exhibit several phenotypes despite the underlying degenerative process being both progressive and relentless (*Skalny et al., 2024*). Despite significant advancements in managing these manifestations, the underlying molecular mechanisms of the disease remain inadequately understood, thereby limiting the potential for identifying effective pharmacological therapies to restore tissue homeostasis (*Umoh, Dos Reis & De Oliveira, 2024*). Such a targeted approach would be particularly beneficial if implemented promptly to mitigate the damage caused by the disease's degenerative and progressive nature (*Huang & Wu, 2018*; *Jansen & Mastbergen, 2022*; *Saito & Tanaka, 2017*). Our research aims to contribute to this initiative by isolating PCD-associated genes from multiple OA-related datasets.

The utilization of machine learning in predicting patient illness is increasingly common (*Swanson et al., 2023*; *Xiong et al., 2023*). However, maintaining accuracy while integrating these methods into clinical practice remains challenging. Critical questions encompass the rationale for selecting specific algorithms and determining the optimal solution. Researcher bias can affect algorithm selection, underscoring the importance of balancing expertise with objective evaluation to minimize bias and maximize insights. By conducting a comprehensive analysis involving GEO and literature screening of genes associated with OA and cell death, AUCell assessment of PCD expression in single-cell transcriptome datasets, and WGCNA to identify intersecting genes, followed by multiple machine learning model analyses, we identified four hub genes (S100A9, PMAIP1, EDA2R, and FASN) associated with PCD. The GO analysis revealed processes including ossification, amplification of cell

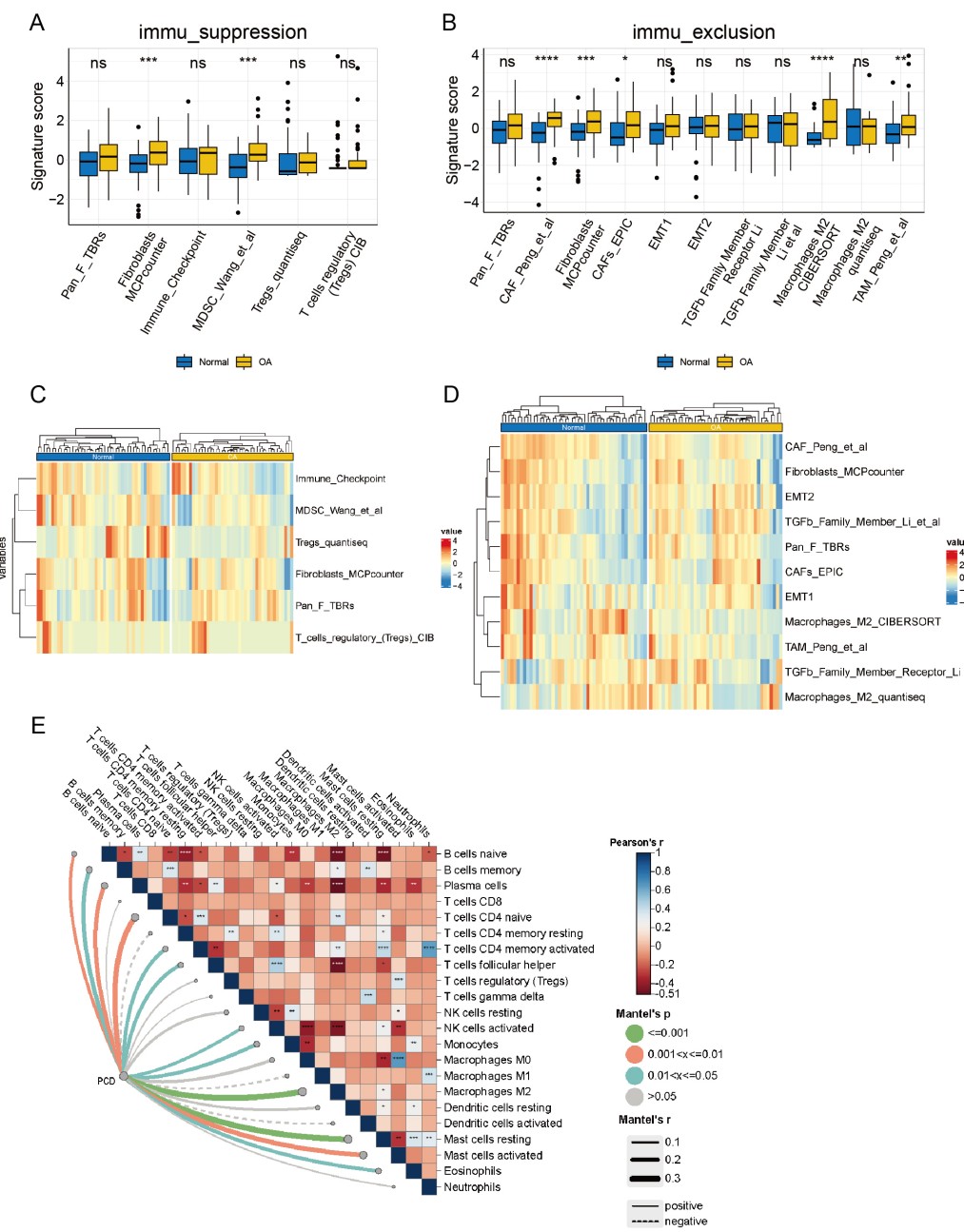

**Figure 9 Analysis of the immune microenvironment from an immunological perspective.** (A) box plot of signature scores for immunosuppression-related features demonstrates the disparity in immunosuppressive cells or pathways between the OA and normal groups. (B) The heatmaps of immunosuppression-related cells and pathways in the OA and normal groups demonstrate specific immunosuppressive characteristics. (C) The box plot of signature scores for immune rejection-related features illustrates the discrepancy in immune rejection characteristics between OA and normal groups, encompassing T cell infiltration, immune escape mechanisms, and other aspects. (D) The heatmap of immune rejection-related features depicts the differential expression of diverse immune rejection features in the OA and normal group. (E) A Mantel correlation heat map was constructed to visualize the intricate web of relationships connecting immune infiltration and PCD, revealing their potential interdependence.

death mechanisms, greater immune cell engagement, intensified regulation of apoptotic signaling, increased neutrophil activation, and enhanced modulation of immunological responses. The KEGG analysis revealed enrichment in pathways, including apoptosis, lysosomes and IL-17 signaling, which are classical pathways of PCD (*D'Arcy, 2019*; *Kopeina & Zhivotovsky, 2022*; *Liu et al., 2022*). Additionally, PCD exhibited a substantial correlation with various immune infiltrations, including M2 macrophages, resting mast cells, and potentially active mast cells, which may be involved in the PCD process.

S100A9, PMAIP1, and EDA2R expression levels were elevated in OA samples, indicating these genes as risk factors for OA. Conversely, FASN expression was decreased in OA samples compared to normal groups, suggesting its function as a protective gene in OA. Single-cell analysis identified anoikis, apoptosis, autophagy, ferroptosis, necroptosis, NETosis, paratosis, and pyroptosis as the predominant active pathways of PCD in OA. Additionally, the development of an *in vitro* cellular OA model through hypoxic treatment of ATDC5 chondrocyte cells revealed abnormal expression patterns that were consistent with bioinformatics analysis. Functional experiments demonstrated that S100A9, PMAIP1, and EDA2R downregulation through siRNA, along with the overexpression of the FASN gene through plasmid transfection, significantly mitigated hypoxia-induced reductions in cell viability, decreased hyaluronan secretion, and increased secretion of inflammatory cytokines (TNF-α and IL-6). These findings indicate that these PCD-related hub genes are essential in OA onset and progression, offering new biological targets for future diagnostic and therapeutic strategies for OA.

Our enrichment analysis revealed a complex interaction, with PCD-related genes predominantly engaged in diverse cellular processes, environmental information processing, and biological systems. This study identified significant associations between OA and immune regulation and a range of metabolic biological processes. These findings may partially clarify the poorer prognosis observed in this group. A key limitation of this study is the lack of *in vitro* or *in vivo* experiments to conclusively validate our findings. Despite utilizing stringent bioinformatics analyses and computational techniques, experimental validation remains essential in scientific research. Such investigation can clarify the functional implications of the observed patterns, hence improving the reliability and robustness of our results. Accordingly, future research should prioritize conducting precise experiments to validate our PCD findings with empirical evidence.

This study has some limitations. First, although gene expression data were obtained from the GEO database, enhancing the accuracy of our findings requires the acquisition of additional samples from clinical patients with OA. Furthermore, the exact relationships within mRNA-miRNA and mRNA-TF networks warrant further investigation. Additionally, the mechanisms of action of the four key genes in OA require more comprehensive investigation through animal experiments and clinical sample analysis. Consequently, these limitations should be considered in future research studies. If feasible, we will undertake experimental validation simultaneously with the collection of clinical samples.

# CONCLUSIONS

Four hub genes associated with PCD, PMAIP1, EDA2R, S100A9, and FASN, were found alongside immune-related cells using machine learning and single-cell analysis. The analysis of competing endogenous RNA networks associated with these hub genes can identify corresponding miRNAs and transcription factors, thereby facilitating the investigation of the underlying mechanisms of OA. Moreover, these key genes provide potential targets for OA immunotherapy. This study provides a new perspective on the complex interactions between PCD-related genes and OA, therefore paving the way for research and therapeutic advancements.

## Funding

The authors received no funding for this work.

## Competing Interests

The authors declare there are no competing interests.

## Author Contributions

- Qinchao Sun performed the experiments, analyzed the data, prepared figures and/or tables, authored or reviewed drafts of the article, and approved the final draft.
- Ye Zhong analyzed the data, prepared figures and/or tables, and approved the final draft.
- Gaoxiang Huang performed the experiments, authored or reviewed drafts of the article, and approved the final draft.
- Yongpei Lin conceived and designed the experiments, performed the experiments, authored or reviewed drafts of the article, and approved the final draft.

## Data Availability

The raw data is available at GenBank: GSE89408, GSE114007, and GSE152805.

Data is available in the Supplemental Files.

## Supplemental Information

Supplemental information for this article can be found online at http://dx.doi.org/10.7717/peerj.20104#supplemental-information.

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
