# Peer review of "Characterizing programmed cell death features in osteoarthritis through integrative multiomics and machine learning analysis"

_PeerJ, doi:10.7717/peerj.20104_

## Round 0.1 · original submission · Major Revisions

**Language Note:** The review process has identified that the English language must be improved. PeerJ can provide language editing services - please contact us at [email protected] for pricing (be sure to provide your manuscript number and title). Alternatively, you should make your own arrangements to improve the language quality and provide details in your response letter. – PeerJ Staff

Reviewer 1 ·

Basic reporting

- In vitro OA model using hypoxiac treatment of ATDC5 chondrocyte cells
- Rationale for this model?
- Clinical translatability?
- Elevated expression levels of S100A9, PMAIP1, and EDA2R suggesting these may serve as risk factors
- Is there literature from humans to support this suggestion?
- FASN reduced in OA - suggesting protective role
- Is there literature from humans to support this suggestion?
- PCD emerged as a reliable diagnostic marker with enhanced predictive accuracy
- Is there literature from humans to support this suggestion?
- What prior work has been done on PCD in OA?

Experimental design

- In vitro OA model using hypoxiac treatment of ATDC5 chondrocyte cells
- Rationale for this model?
- Clinical translatability?
- What prior work has been done on PCD in OA?
- Clinical data and transcriptomic profiles of individuals with OA obtained from GEO (ncbi) repositories
- 89 samples with 50 selected from GSE89408 and 49 from GSE114007
- What are each of these?
- Useful summarization of PCD modalities and the number of genes that contribute to each
- “In our analysis of genes with significant expression differences between the highest and lowest PCDI classes, we adhered to thresholds of FDR less than 0.05 and log2 fold change exceeding one.” (Cite?)
- (Line 165) “To ensure the robustness of our analysis, we carefully removed low-quality cells. These included cells with gene expression detected in three or fewer cells, those with fewer than 200 detectable genes, cells exhibiting a mitochondrial gene ratio above 15%, a ribosomal gene ratio below 3, and those with nFeature counts exceeding 7500.” (Cite?)
- (Line 216) “Predictive models were developed using the GSE89408 dataset and subsequently validated across two independent external datasets (GSE55235 and GSE114007)” (rationale for the one chosen as training and the two chosen as validated?)
- (Line 253) “By utilizing ³-actin as a reliable internal control gene, we effectively mitigated variability among samples, allowing for precise evaluation of target gene expression changes under various treatment conditions” (Cite?)
- Re: statistical analyses section: How was correction for multiple hypothesis testing conducted?

Validity of the findings

Results
- Figure 1A-B would benefit from a caption underneath each indicating that 1a is before removal of batch effects and 1b is after removal
- Figure 1E how do you interpret staph aureus as the number one more differential pathway between the two groups? And its implications to the study / findings more broadly? Also another infection being the #4?
- Figure 2B what does one vs two vs three vs four asterisks mean? Level of significance? Figure would benefit from including a legend re: number of stars
- How do you interpret these results?
- Figure 3E is not mentioned in the results. There is no description of how 3D was collapsed into 3E? Moreover, there is considerable heterogeneity between individual clusters in 3D. How can you be confident in the cell types assigned?
- Figure 6B-C, (line 377) the AUC are referred to as strong (in the 0.70 range). Generally 0.70 - 0.80 is considered acceptable whereas 0.80 - 0.90 is considered “strong” or “excellent” (citation: 10.1097/JTO.0b013e3181ec173d)
- Figure 7B (line 388) the AUC of the predictive potential of the HUB genes was referred to as “high”. I’d lighten to “significant” as literature seems to consider >0.8 as “high” (10.1097/JTO.0b013e3181ec173d)
- What does the literature say about these genes and OA?
- Cite line 397-398?
- Figure 8 - this figure really brings it home and demonstrates the value of transcriptomics, single cell sequencing, and machine learning methods to identify potential targets for ameliorating disease. Not necessary for this manuscript, but curious if the authors tried combining any of the changes made to HUB genes to measure additive vs. synergistic effects?
- (Line 424) - MDSC_Wang_et_al and Fibroblasts MCPcounter are not explained in the methods same with CAF_Peng_et_al, Fibroblasts MCPcounter, CAFs_EPIC, Macrophages M2 CIBERSORT, and TAM_Peng_et_al (lines 419 - 420)

Additional comments

This manuscript demonstrates the value of transcriptomics, single cell sequencing, and machine learning methods to identify potential targets for ameliorating disease in the context of osteoarthritis.

The manuscript title could potentially be leveled up to:

Identifying potential pathologic and protective genetic features related to programmed cell death in osteoarthritis through integrative multi-omics and machine learning analysis

Reviewer 2 ·

Basic reporting

All comments have been added in detail to the last section.

Experimental design

All comments have been added in detail to the last section.

Validity of the findings

All comments have been added in detail to the last section.

Additional comments

Review Report for PeerJ
(Characterizing Programmed cell death features in osteoarthritis through integrative multi-omics and machine learning analysis)

1. This study reveals the diagnostic, genetic, and therapeutic significance of programmed cell death (PCD)-related genes in osteoarthritis through multi-omics analyses and AI-based approaches, offering new targets for personalized treatment strategies.

2. In the introduction, what osteoarthritis is, the importance of the subject, and the literature are discussed at a basic level. In this section, it is suggested that a literature table consisting of certain columns such as "pros and cons, method, results" be added in order to express the literature more clearly. After this, the main contributions of the study to the literature should be stated more clearly and in bullet points.

3. When the type and amount of dataset used in the study is examined in terms of the problem addressed in the study, it is observed that it is at a sufficient level.

4. Examining the Box Plots, Dot Plots and Heat Maps of the differential expression of PCD-related genes in OA, the nature and importance of the study becomes clearer.

5. When the relevant figures and results are examined in relation to Evaluating the ability of genes by combining multiple machine learning models, it is observed that certain metrics are obtained in which a significantly different machine learning model is preferred compared to the literature. However, in this section, obtaining some missing metrics such as Matthews correlation coefficient (MCC) and Cohens Cappa will further increase the confidence and quality of the results of the study.

The study has the potential to make significant contributions to the literature. However, the sections listed above should be noted.

---

## Round 0.2 · accepted · Accept

It appears to me you have addressed the reviewers' comments quite well. Thus, I recommend "accept".